# The Complex Biology of the Obesity-Induced, Metastasis-Promoting Tumor Microenvironment in Breast Cancer

**DOI:** 10.3390/ijms23052480

**Published:** 2022-02-24

**Authors:** Noshin Mubtasim, Naima Moustaid-Moussa, Lauren Gollahon

**Affiliations:** 1Department of Biological Sciences, Texas Tech University, 2500 Broadway, Lubbock, TX 79409, USA; nmubtasim.mubtasim@ttu.edu; 2Department of Nutritional Sciences, Texas Tech University, 2500 Broadway, Lubbock, TX 79409, USA; naima.moustaid-moussa@ttu.edu; 3Obesity Research Center, Texas Tech University, 2500 Broadway, Lubbock, TX 79409, USA

**Keywords:** obesity, white adipose tissue, breast cancer, migration, invasion, breast tumor microenvironment, proinflammation

## Abstract

Breast cancer is one of the most prevalent cancers in women contributing to cancer-related death in the advanced world. Apart from the menopausal status, the trigger for developing breast cancer may vary widely from race to lifestyle factors. Epidemiological studies refer to obesity-associated metabolic changes as a critical risk factor behind the progression of breast cancer. The plethora of signals arising due to obesity-induced changes in adipocytes present in breast tumor microenvironment, significantly affect the behavior of adjacent breast cells. Adipocytes from white adipose tissue are currently recognized as an active endocrine organ secreting different bioactive compounds. However, due to excess energy intake and increased fat accumulation, there are morphological followed by secretory changes in adipocytes, which make the breast microenvironment proinflammatory. This proinflammatory milieu not only increases the risk of breast cancer development through hormone conversion, but it also plays a role in breast cancer progression through the activation of effector proteins responsible for the biological phenomenon of metastasis. The aim of this review is to present a comprehensive picture of the complex biology of obesity-induced changes in white adipocytes and demonstrate the relationship between obesity and breast cancer progression to metastasis.

## 1. Introduction

Breast cancer is the leading cause of the cancer-related deaths in women. It is the most common cancer prevalent globally surpassing lung cancer for the first time in 2020 [1]. Additionally, in 2020, more than 2.3 million women were diagnosed with, and 685,000 women died from breast cancer worldwide [2]. It is the second most prevalent cancer after lung cancer and the most diagnosed disease among U.S. women [3]. One in every eight women has the lifetime risk of developing breast cancer in the United States [4]. Until the end of 2018, there were about 3.8 million women with a history of breast cancer in the United States [5]. According to the latest update by the American Cancer Society on breast cancer statistics, it was estimated that there would be 281,550 new cases of invasive breast cancer in women in the U.S. and 43,600 women would die from the disease by the end of 2021 [6]. With the advancements in early cancer detection, easy access to screening, and improved treatment options, the death rate of breast cancer has decreased by 40% between 1989 and 2017 [6]. However, it is still a major health issue in the advanced world, primarily because 90% of breast cancer-related deaths are attributed to metastasis [7]. The 2017 report from the National Cancer Institute suggests that more than 150,000 women from every ethnicity and race suffer from breast cancer metastasis in the U.S.A. [8]. This estimate highlights the urgency of identifying the potential risk factors behind the poor prognosis of breast cancer. Further, correlating these risk factors with the disease biology from a pathological point of view yields important information to better combat the mortality rate due to metastasis. In this review, we take a comprehensive look into the complex biology of obesity-induced changes in white adipocytes that promote a metastatic phenotype in breast cancer.

## 2. Breast Cancer

### 2.1. Molecular Histotypes of Breast Cancer

Breast cancer is a highly heterogeneous disease. The aggressiveness of breast cancer varies according to its molecular subtypes. The different subtypes are derived from different cells of origin within the mammary glands [9]. They are classified into the following subtypes, luminal A, luminal B, HER2 positive, and basal type or triple negative breast cancer (TNBC), based on the presence of key protein receptors, namely estrogen receptor (ER), progesterone receptor (PR), and human epidermal growth factor receptor 2 (HER2) [10]. Each subtype has its own distinct set of risk factors, presentation, prognosis, and response to treatment (Table 1). TNBC (ER-/PR-/HER2-) has the worst prognosis of all the types and has the highest metastatic potential [10]. The treatment outcome of TNBC is also very poor as tumor cells of this subtype do not express any hormone/growth factor sensitive receptors, such as ER, PR, and HER2. On the other hand, luminal A, luminal B, and HER2 positive tumors all exhibit some level of response to treatment due to the presence of hormone sensitive receptors on their surface (Table 1). Inflammatory breast cancer (IBC) is another aggressive and rare form of breast cancer with poor prognosis. Epidemiological studies reported that the majority of the cases of IBC are hormone receptor negative and have a preponderance of HER2+ overexpression [11]. Clinically, IBC can be defined as breast cancer with a rapid onset of inflammatory-like symptoms, such as edema, redness, body warmth, and skin dimpling, due to the formation of tumor emboli in the papillary and reticular dermis layer of the breast [11]. This tumor was identified based on its molecular characterization. The molecular profiling of IBC confirmed mutations in MYC, TP53, components of the phosphoinositide-3-kinase (PI3K) pathway (AKT1, AKT3, PTEN, and PIK3CA), and the overexpression of the oncogene Rho C [11].

### 2.2. Etiology and Ethnic Differences in Breast Cancer

In the biennial report by the American Cancer Society, certain factors were listed as contributing towards elevating the risk of breast cancer development. The inherited variation in the genetic sequence of BRCA1, BRCA2 genes accounts for 5% to 10% of all breast cancers and was identified primarily among Black and Hispanic breast cancer patients [13]. Other factors that contribute to the risk of developing invasive breast cancer include the use of hormonal contraceptives, menopausal hormone replacement therapy, nulliparity, childbirth at a late age, lack of breastfeeding, early menarche, and late menopause [13]. In addition, the increase in body mass index (BMI) due to changes in dietary behavior and physical inactivity also progressively aggravates the overall survival rates [13]. Excess weight gain and its likelihood for fueling the pathogenesis of breast cancer is mainly devastating for women who are above age 50 or who are postmenopausal [13,14]. This is because fat cells become the major source of estrogen post-menopause, subsequently increasing the risk of developing hormone-sensitive breast cancer in obese women [13]. Aging is another important variable that increases the susceptibility of breast cancer development. The most invasive breast cancers are reported in women who are >55 [15].

Some ethnic groups are more affected by the risks associated with breast cancer. According to a report by Daly et al. [16], while Caucasian women have the highest incidence of reported breast cancers, African American women are more likely to die from the disease. Many factors can contribute to these differences in survival rates. The histotype of the breast tumor is one of them. Caucasian women suffer primarily from hormone receptor-positive breast cancer, which has greater treatment success due to the available receptor-targeting medications. Furthermore, hormone receptor-positive tumors are very slow growing and have less chances for recurrence. However, Black, non-Hispanic Black, and African American women are more susceptible to developing hormone receptor-negative or triple-negative breast cancers, which are more aggressive. These types of tumors are very difficult to treat via hormone therapy or HER2-targeted drugs and often become resistant to chemotherapy. As a nation of diverse populations, the United States of America has the highest prevalence of breast cancers with different prognoses, and necessarily requires varied treatment strategies. Thus, there is a continuing need to understand breast cancer etiology better for the effective treatment outcome across diverse populations of women in the U.S.

### 2.3. The Tumor Microenviroment

Several experimental studies suggest that the tumor microenvironment (TME) plays a role in determining the prognosis of breast cancer. Autocrine and paracrine signaling factors from the TME primarily stimulate the growth and proliferation of cancer cells during the earlier stage of tumorigenesis [17]. Invasive breast cancer usually presents as a solitary tumor. However, it has the distinct ability to spread to nearby organs as well as to distant areas across the body. This complex process of cancer spreading from its original, or primary, location to new areas is known as “secondary cancer” or “metastasis” [18]. Metastasis is one of the major reasons why breast cancer is still incurable [18]. It is a coordinated multi-step process involving the dissociation of breast tumor cells from their original location, followed by their endothelial transmigration, and eventual colonization in the competent organ [19,20]. Scientists have long tried to answer the question: “what makes a tumor cell metastatic?”. Previously, it was believed that the presence of cellular heterogeneity within the tumor population worked as a driving force for metastasis [21]. This heterogeneity is aided by epigenetic instability, genetic alterations caused by DNA mutations, and chromosomal remodeling [22]. However, recent evidence demonstrated that in response to the changing physiological conditions, the TME is much more dynamic than previously thought and can contribute to initiating the metastatic cascade [21].

Among the modifiable risk factors, obesity is the most alarming health issue, which increased the risk of several cancers, including breast cancer, by being a contributor to changing the normal homeostasis of the TME. It is reported to be responsible for 52% and 88% cancer-related mortality rates in males and females, respectively [23]. The strong association of obesity with several pathological conditions, such as type 2 diabetes, cardiovascular disease, and hypercholesterolemia, is well established. However, their link with cancer is still underappreciated [24]. Breast tissue is distinctive in nature due to the presence of a fat-rich connective tissue known as the mammary fat pad, of which white adipocytes constitute its major cellular component [25]. Epidemiological studies reported that obesity-associated changes in the metabolic profile and pro-inflammatory signaling of the adipose tissue disrupts the normal physiological homeostasis in local tissues and the systemic microenvironment. This, in conjunction with hypoxia, creates a critical modification in the TME that increases the risk of both the progression and pathogenesis of breast cancer in obese individuals [24]. With the increased dependence on a materialistic lifestyle, less physical activity, and laborious work, coupled with the overconsumption of energy-dense foods, the U.S. in now the global leader for having the highest obesity rates [26]. According to the State of Obesity: Better Policies for a Healthier America 2018, the obesity rate reached 40% in 2018 after staying ~34–35% between 2005 and 2012 [27]. In 2020, it increased to 42.4% with a higher rate in women than men [28]. The continuously increasing rates of obesity have put millions of lives of women that has developing postmenopausal breast cancer at risk [28].

### 2.4. Chronic Inflammation and Breast Cancer

As early as 1863, Rudolf Virchow was the first to hypothesize a link between chronic inflammation and cancer [17]. Obesity promotion, chronic inflammation, and endocrine changes in adipose tissue, act as intermediaries in building a “rapport” between obesity and cancer [29]. Adipose tissue is a non-trivial part of the breast fat pad where white adipose tissue (WAT) comprises the majority of the deposits [2]. WAT is heterogenic in nature and consists of a complex cellular arrangement that makes the TME dynamic in nature. The main function of WAT is to store fat in the form of triglycerides and release it in the form of fatty acids whenever there is a demand for energy or other metabolic needs. It is also an active endocrine organ that secretes more than 50 kinds of bioactive compounds [30]. Under normal physiological conditions, the breast’s luminal epithelial cells require an interaction with the mammary fat pad for ductal luminal cell morphogenesis and alveolar luminal cell differentiation into lactating alveoli [25]. Moreover, the presence of immune cells in healthy breast tissue maintains tissue homeostasis and an immune suppressive microenvironment by keeping track of apoptotic cells, angiogenesis, and extracellular matrix (ECM) remodeling [31]. However, due to increased fat accumulation as a result of obesity, white adipocytes undergo an increase in volume [32]. Adipocyte hypertrophy compromises the integrity of some adipocytes due to limited expandability. This overaccumulation of fat disrupts metabolic homeostasis and induces cell stresses, such as ER stress and oxidative stress, resulting in a chronic inflammatory microenvironment [30,32,33]. Such deregulation assists in facilitating different obesity-associated diseases, including breast cancer [14].

Obesity-induced chronic inflammation stimulates the production of reactive oxygen species (ROS) and reactive nitrogen species (RNS) [34]. This ensures tumorigenesis by inducing genetic mutations that activate protooncogenes, chromosome alterations, and the inactivation of tumor suppressor genes [34]. Inflammation also causes the infiltration of proinflammatory macrophages in response to chemotactic agents, which further secretes proinflammatory cytokines and chemokines in to the TME [35]. This, in turn, induces the activation of transcription factors, namely, the nuclear factor kappa-light-chain-enhancer (NF-kB), signal transducer, and activator of transcription factor 3 (STAT-3), signal transducer and activator of transcription factor 5 (STAT-5), and hypoxia inducible factor 1-alpha (HIF-1a), which activates signaling pathways progressing towards more aggressive breast cancer, and increasing the likelihood of metastasis [36,37]. In short, obesity-induced changes in adipocytes, followed by the recruitment of macrophages, results in a chronic inflammatory condition in the breast TME. This condition causes changes in the secretory profiles of adipocytes around breast cells. These molecules further play a role as mutagens or signaling molecules controlling the carcinogenic or metastatic events in breast epithelial cells. Despite being a prominent cell-type present in the mammary stroma, adipocyte tissue biology is less studied and investigated. This review focuses on how obesity-associated changes in white adipocytes and TME play a conducive role towards the progression of breast cancer by rewiring its cell-signaling pathways towards migration and invasion. Understanding the underlying pathogenic changes in white adipocytes and breast cancer cells in obesity helps us to understand the relationship between obesity and breast cancer metastasis better.

## 3. Obesity, Inflammation, and Breast Cancer—A Vicious Cycle

### 3.1. Obesity-Associated Chronic Inflammation

Obesity-induced changes in WAT reprograms the TME in such a way that it becomes favorable for the growth and progression of breast cancer. Among the numerous mechanisms behind the risks associated with obesity-induced breast cancer, the role played by inflammation remains arguably at the forefront. Due to excess fat accumulation by WAT and the subsequent reduction in blood supply resulting in regional hypoxia and adipocyte stress [24,38], reduced tissue oxygenation causes changes in the transcriptional programming of adipocytes and other stromal cells, which leads to the excess deposition of fibrillar components in the ECM [2]. This makes the ECM stiff and generates tissue fibrosis [2]. Adipocytes, enclosed within this rigid ECM, face metabolic perturbations, which then impair their normal physiological functions and entail necrotic or pyroptosis-induced cell death [2]. Adipocyte cell death, due to a rupture in the cell membranes, accompanies the release of cellular contents, such as lipids, cytokines, ROS, RNS, the nucleic acid [24]. These adipocyte-released contents attract immune cells to the dying adipocytes [39]. Increased fatty acids released from the dysfunctional adipocytes are recognized by the Toll-like receptor 4 (TLR4) present on the cell surface of the resident macrophages [35,40]. This leads to the overproduction of proinflammatory cytokines via NF-kB activation, followed by the creation of a localized inflammation in the breast TME (Figure 1) [35,38].

Inflammation is further intensified by the recruitment of macrophages. Obese dysfunctional adipocytes induce localized inflammation, increasing the level of monocyte chemoattractant protein (CCL2) expression by adipocytes in the breast TME (Figure 1). Additionally, cell-free DNA, released from the degenerated adipocytes, also increases the level of CCL2 protein [41]. Several studies reported CCL2 as an independent predictor of recruited macrophage accumulation in the adipocytes. One in vivo study showed how CCL2/CCR2 knockout mice show reduced infiltration of macrophages [42]. Such relevant in vivo studies also delineated the fact that a high-fat diet may be an influencer for the increased level of CCL2 proteins in those mice [42,43]. CCL2 works as a chemoattractant, recruiting peripheral blood monocytes bearing the chemokine receptor CCR2, from the surrounding intravascular spaces to the TME [41]. Under environmental cues, these recruited monocytes differentiate into macrophages [44]. Tumor-derived colony stimulating factor 1 (CSF-1) also attracts monocytes in a paracrine fashion to the tumor site [45]. Hence, there persists a sustained accumulation of recruited monocytes in breast TME [44]. Macrophages gather around the dying adipocytes in a crown-like structure and phagocytose them. Entrapped lipid-loaded macrophages produce ROS and RNS, which further act as mutagens [25,41]. These changes cause increased intracellular signaling in macrophages via nuclear factor kB (NF-kb), signal transducer, and activator of transcription 3 (STAT3), c-jun-NH2 terminal protein kinase (JNK)-related pathways, followed by the release of proinflammatory cytokines [31,41]. Such prolonged and sustained proinflammatory signaling creates a state of chronic inflammatory breast TME (Figure 1). Thus, macrophage-derived cytokines create a chronic low-grade inflammatory state that works as a mediator in building a bridge between obesity and breast cancer pathogenesis.

### 3.2. Obesity-Associated Macrophage Corruption

The process of macrophage infiltration is positively associated with increased body weight and adipocyte size [46]. The enrichment of macrophages at the invasive front of the primary tumor site is reported in [44]. One in vivo study demonstrated the importance of CSF-1, a nutrient for macrophage survival and proliferation, by correlating its deletion with reduced macrophage density and delayed the metastatic progression of cancer [47]. Based on these experimental results, macrophages, recruited in the tumor microenvironment, are indirectly associated with tumor progression, and are identified as tumor-associated macrophages (TAMs) [48,49]. TAMs are responsible for breast tumor progression by fostering EMT, angiogenesis, extracellular matrix degradation, immunosuppression in response to the TME [49]. In addition to the recruited macrophages, there are also tissue-resident macrophages present in the breast TME [49]. However, they play contrasting roles. Tissue-resident macrophages are derived from hematopoietic progenitors present in the sac, which arises during embryonic development [50]. Colony stimulant factor 1 (CSF1) produced by local stroma maintains the in situ proliferation and self-renewal of local macrophages [50]. Under normal physiological conditions, they maintain tissue homeostasis and defend against pathogens [44].

Due to their high plasticity, macrophages can transform into different endotypes based on the cues released from the TME [50]. While classically activated tumor-associated macrophages (M1) restrain cancer development, alternatively activated tumor-associated macrophages (M2) promote cancer [48]. The hypoxic breast tumor microenvironment is very conducive for M2 polarization of tumor-associated macrophages [41]. An in vivo study reported that the low level of expression of the EMT transcription factor, SNAIL1 in tumor cells, induced the M1 polarization of macrophages, whereas the higher level of expression of SNAIL1 in tumor cells elicited M2 polarization by SNAIL1-mediated production of the granulocyte-macrophage colony-stimulating factor (GM-CSF) [51]. This study suggests that the increased expression of SNAIL1 by invasive mammary tumor cells, increases the probability of recruited macrophages polarizing towards the M2 phenotype [51]. The presence of these alternatively activated macrophages at tumor-stroma sites play an important role in connecting obesity-induced inflammation with cancer [41].

### 3.3. Obesity-Associated Aromatase Expression

Apart from the interaction between adipocytes and macrophages in the breast TME, the availability of estrogen also increases, especially in postmenopausal breast cancer patients. Studies have observed a high prevalence of obesity correlated with increased serum concentrations of estrogen in postmenopausal breast cancer patients [46]. Estrogen is a sex steroid hormone that is mainly associated with hormone receptor-positive breast cancer. This hormone is primarily produced in the granulosa cells of the ovaries in premenopausal women [46]. Under normal physiological conditions, white adipocytes are also capable of producing sex steroid hormones. As a result, in postmenopausal women, when the ovaries stop producing estrogen, white adipocytes become the main source for its biosynthesis [31]. A total of 18 carbon steroid estrogens (e.g., estrone and estradiol) are mainly converted from 19 carbon steroids androgens (e.g., androstenedione and testosterone) in the presence of aromatase (Figure 1) [46,52]. The obesity-associated increased release of proinflammatory cytokines in breast TME is accompanied by an increased expression of aromatase in white adipocytes, which then converts androgens to estrogens in the adipose tissue [46]. In addition to this, obesity-assisted hyperinsulinemia decreases the level of steroid sex hormone-binding globulin (SHBG) [53]. This increases the bioavailability of free estrogen, which then promotes mammary tumorigenesis (Figure 1). In addition to an increased BMI, women with increased mammographic density have also been found to be positively correlated with an increased activity of aromatase, and, consequently, increased risk of breast cancer [46].

Several studies tried to explain how the chronic inflammatory condition controls the expression of aromatase. In one review, Wang suggested that, generally, the expression of aromatase is encoded by the human gene CYP19A1, under the regulation of the tissue-specific promoters PI.4, PII, and PI.3 [46]. These promoters are responsible for the mRNA expression of aromatase in adipose tissues. Although the promoters remain in low abundance under basal conditions, their activity increases in cancer. High concentrations of the promoters were positively correlated with breast cancer progression. Obesity-associated inflammatory mediators, TNF-α, IL-6, together with glucocorticoids, stimulate the activity of PI.4, which subsequently increases the expression of aromatase. The formation of dense fibroblast layers surrounding the malignant cells due to the TNFα-induced dedifferentiation of adipocyte stromal cells also causes an enhanced expression of aromatase. Simultaneously, chronic inflammatory tumor macrophage-derived factor prostaglandin E2 (PGE_2_) stimulates the activation of PI.3/II promoters via MAPK/JNK-mediated pathways and further enhances the expression of aromatase [46]. One study found that hypoxia inducible factor (HIF1α) is also involved in the expression of aromatase. In addition, insulin-like growth factor-1 (IGF-1), another adipocyte-secreted factor present in hyper insulinemic obese patients, can stimulate the activity of aromatase in adipose stromal cells [54]. In summary, obesity-related changes in the circulating levels of adipokines and inflammatory mediators orchestrates a microenvironment permissive for the growth of hormone-sensitive ER+ breast cancer [29,55,56,57].

### 3.4. Obesity-Associated Metabolic Remodeling

One of the several benefits of breast tumor cells having adipocytes in the microenvironment is the mobilization of stored fat for compensating higher metabolic requirements [25]. Cancer cells require constant supplies of nutrients for their metabolic growth and energy. Various in vitro and in vivo studies concluded that cancer cells offset this need by generating de novo synthetic pathways to generate fatty acids from blood-derived glucose (Figure 2) [58,59]. The de novo biogenesis of fatty acid (FA) and cholesterol, under normoxic conditions, takes place through the conversion of glucose to pyruvate, which feeds into the tricarboxylic acid (TCA) cycle and generates citrate [58,60]. This mitochondrially produced citrate is converted to acetyl Co A, a substrate of fatty acid synthesis, with the help of ATP citrate lyase [61]. One molecule of Acetyl Co A, with seven molecules of malonyl Co A, in the presence of the fatty acid synthase (FASN) and reduced equivalent NADPH, generates the initial FA palmitate via serial condensation [59,60]. This saturated long chain fatty acid is then further modified and desaturates into complex membrane phospholipids and triglycerides (TAGs) along the way, due to the action of different enzymes [58]. In vivo and in situ studies reported an increased expression of FASN in breast cancer cells [62,63]. FASN-catalyzed FA biosynthesis is stimulated by a high-carbohydrate diet and is suppressed during fasting. Acetyl CoA is also the initial substrate for cholesterol synthesis following the mevalonate pathway. The sterol regulatory element-binding proteins (SREBPs), the master regulator or transcription factor controlling the expression of enzymes involved in fatty acid and cholesterol synthesis, is also overexpressed in cancer [59,60].

However, this fatty acid can also be supplied by the TME. In a proliferative TME, a lack of oxygen due to insufficient diffusion to an area distant from vasculature, causes hypoxia [64]. Hypoxia triggers cancer cells to compete with the resident stromal and immune cells for metabolic nutrients [65]. Under such conditions, cancer cells use aerobic glycolysis, where glucose-derived pyruvate is converted to lactate to generate ATP instead of using TCA cycle facilitated mitochondrial oxidative phosphorylation (Figure 2) [65]. To compensate for the need of fatty acids, cells may switch to alternative sources, such as acetate or the glutamine pathway, to biosynthesize lipids [59]. However, in hypoxic conditions, the synthesis of monounsaturated fatty acids is compromised due to oxygen limitations, compromising the enzymatic reactions [66]. The unavailability of unsaturated fatty acids, in turn, raises endoplasmic reticulum stress and triggers the activation of the uncoupled protein response (UPR), due to the over-incorporation of saturated fatty acids into the ER membrane, leading to cell death [66]. To survive against these oxygen and nutrient constraints, cancer cells metabolically depend upon the microenvironment. This is where adipocytes come into play, and this interaction has severe consequences in the case of obesity.

The rapid hypertrophy of white adipocytes due to excess caloric intake coupled with a lack of energy expenditure results in increased fat storage [67]. Because free FAs are toxic at higher concentrations, they are stored in the cytosolic lipid droplets of white adipocytes in the form of triglycerides [64]. Triglyceride synthesis and storage in adipocytes takes place through the glycerol–phosphate pathway (Figure 2) [23]. Free FAs, in the presence of acyl CoA synthetase, forms 2 molecules of fatty acyl CoA. This fatty acyl CoA is acylated and dephosphorylated to diacyl glycerol in a reaction with glycerol-3-phosphate [23]. Triglycerides are synthesized when a third molecules of fatty acyl CoA is added to the glycerol backbone in the presence of diacylglycerol transferase [23]. Moreover, in the obese condition, the expression of FASN also increases in adipose tissues, thus influencing the de novo lipogenesis of triglycerides and their storage in adipocytes [68]. This increased synthesis of triglycerides from free FAs and their accumulation as lipid droplets, results in adipocyte hypertrophy leading to obesity.

Although the average range of lipid droplets in other cells is 0.1 to 10 μm, in adipocytes they can accumulate up to 100 μm [64]. Due to the proximity, breast cancer cells use lipids from adipocytes as a source of energy [67]. Adipocytes can mobilize their stored lipids for breast cancer cell metabolic requirements through neutral lipolysis and autophagy. With the help of the catabolic enzyme adipocyte triglyceride lipase (ATGL), hormone sensitive lipase (HSL), monoacylglycerol lipase (MAGL), triglycerides, or triacylglycerol (TAG) are hydrolyzed into fatty acids (Figure 2) [67]. ATGL converts triglycerides to diacylglycerol. Then, in the presence of HSL, diacylglycerol converts into monoacyl glycerol [23]. In the final step, monoacylglycerol breaks down to a free FA and glycerol via the MAGL enzyme-mediated hydrolysis process [23]. Thus, in obesity, there is an increased availability of circulating free fatty acid for breast cancer cells to offset their continuous metabolic needs. One in vitro study reported that the coculture of breast cancer cells and/or conditioned medium treatment from those cells on mouse 3T3-L1 adipocytes induces lipolysis from their triglyceride stores and increases the release of fatty acids [69]. Released fatty acids are then taken up by the transmembrane channel protein CD36 found to be overexpressed in metastatic cancer cells [64]. CD36, also known as fatty acid translocase (FAT), is from a solute carrier protein family that can bind to free fatty acids and facilitate their efficient movement across the plasma membrane [70]. Further, evidence has also shown that fatty acid-binding protein 4 (FABP-4) is also found to be increased in breast cancer cells and is responsible for the intracellular trafficking of fatty acids [67].

The excess FAs present in the obese and inflamed breast tumor microenvironment benefit cancer cells in many ways. FAs derived from adipocytes are an energy source for breast cancer cells following β-oxidation [25,69]. FAs can produce 3–4 times the amount of energy compared to glucose [59]. Fatty acid oxidation generates NADH, FADH_2_, and acetyl CoA [64]. NADH and FADH_2_ then enters the electron transport chain to generate more energy [64]. One in vitro study demonstrated that triple-negative breast cancer cells maintain high levels of ATP by using the β-oxidation bioenergetic pathway [71]. In addition, released FAs may upregulate proinflammatory cytokines by activating Toll-like receptor 4 on the macrophages, thereby further promoting inflammation [67]. Moreover, FAs are also used to generate bioactive lipid signaling molecules, such as eicosanoids and prostaglandins. These lipid signaling molecules not only play a role in facilitating in situ cell survival, growth, proliferation, and metastasis by activating different oncogenic signaling pathways, but can also facilitate immune evasion [67]. PGE2 induces the conversion of helper T cells to the immunosuppressive type 2 phenotype [67], and thus imparts immunosuppression by downregulating the antigen presentation capability of dendritic cells. They further decrease the viability of natural killer cells, ensuring an immunosuppressive microenvironment conducive for cancer progression [72]. Thus, obesity-associated increased fat accumulation in the mammary fat pad, followed by the increased availability of free fatty acids, makes a favorable microenvironment for the advanced progression of breast cancer.

In addition to making the microenvironment favorable for breast cancer progression, the excess consumption of dietary trans fats in obesity also plays a role in driving the breast cancer cell to a more aggressive and metastatic phenotype. Fatty acids are essential building blocks for membrane lipids, such as phospholipids, glycolipids, and cholesterol [64]. One in situ study reported that that the increased proportion of saturated FA used in building the membranes lipids increases the chances of the survival of breast cancer cells [73]. The greater incorporation of saturated FAs in membrane lipids facilitates breast cancer cells’ survival against oxidative stress because they are less susceptible to peroxidation [67]. Hence, under these circumstances, the proportion of saturated and unsaturated fatty acids in the cell membrane also plays a role in cancer cell survival and progression. Furthermore, lipid rafts are distinct microdomains of membrane lipids with a high cholesterol and sphingolipid content. By housing different membrane lipids, such as G-protein coupled-receptors and tyrosine receptor kinases, they act as a platform for diverse cell-signaling pathways in breast cancer [67]. Lipid rafts also localize drug transporter membrane proteins, which can lead to chemoresistance. This phenomenon can be reversed by incorporating polyunsaturated fatty acids (PUFAs) into membrane lipid components [67]. This disorganizes the lipid raft microdomain composition and their function by opposing the activation of oncogenic signaling pathways involved in cell survival and proliferation [67]. In this way, dietary trans-fat intake affects the cancer cell membrane lipid composition and its physicochemical properties, facilitating a more aggressive tumor phenotype.

### 3.5. Obesity Favors Metastatic Behavior in Cancer Cells

Cell migration is an important fundamental biological response conserved not only in simple unicellular organisms, such as amoebas, but also in multicellular mammals where the process controls numerous biological events, such as embryonic development, wound healing, and immune cells invasion. It is also a significant biological phenomenon in pathological conditions, such as cancer metastasis. Cell migration is a mechanical phenomenon based on the coordination of the membrane protrusion formation, contractile force generation, and cell–matrix adhesion [74]. The complex multistep process of a tumor cell spreading from its original or primary location, by invading the basement membrane towards the blood vessel to colonize a distant location, is known as “secondary cancer” or “metastasis” [18]. The precise coordination and integration of signals from the adjacent cells and extracellular matrix is essential for this purpose [75]. Tumor cells of an epithelial origin break off the primary tumor mass to migrate to distance locations in metastasis. As a result, the attainment of migratory properties with less intracellular connection is needed [76]. The conversion of epithelial cells to the mesenchymal phenotype is an early event towards this multistep process [75]. During this transition, epithelial cells start exhibiting mesenchymal-like characteristics by losing their polarity and cell–cell adhesion, and acquiring the malignant characteristics of cell migration and invasiveness [77].

The epithelial-to-mesenchymal transition (EMT) was found to be aberrantly expressed under pathological conditions, such as cancer [78]. An large number of studies have shown the role of EMT in cancer metastasis [79,80]. The genetic alteration of transcription factors is the core regulator of EMT, causing changes in the expression of effector proteins that control cell adhesion and cell polarity [78,81,82]. Hypoxic conditions in the primary tumor mass aggravates the process through the expression of the hypoxia-inducible factor (HIF-1) [81]. Obesity further exacerbates the cancer condition in patients, as it propels primary tumor cells towards EMT events and facilitates malignant progression [69,83]. One of the hallmarks of EMT is the “cadherin switch”, characterized by a loss of E-cadherin and the increased expression of N-cadherin [84]. At the transcription level, the regulation of EMT is controlled by the transcription factors SNAIL, Twist, and ZEB-1 (zinc finger E-box-binding homobox 1 protein). They bind to the promoter region of EMT-targeted genes that downregulate the gene expression encoding for E-cadherin and increase the gene expression encoding for N-cadherin [85]. A recent in vitro study on colorectal cancer cells presented how the expression of inflammatory cytokines increases EMT [85]. Another in vitro study on breast cancer cells T47D and MCF-7 presented how macrophage-derived conditioned medium induced the expression of EMT biomarker proteins in them [86]. These studies suggest that the inflammatory microenvironment is a potent inducer of EMT. Obesity-associated secretion of growth factors and adipokines from adipocytes was also found to activate EMT-associated genes by inducing the transcription factors Snail1/2, Twist1, and Zeb ½, via he activation of the intracellular kinase cascade [81,87,88,89,90]. This triggered the loss of cell adhesion and junctional protein arrangements, such as occludin, E-cadherin, and claudins, resulting in altered cell polarity towards a spindle-like morphology [82]. This newly formed mesenchymal-like phenotype in breast cancer cells has a migratory behavior. Additionally, the loss of E-cadherin increases the chances of breast cancer cell survival due to a loss of anoikis or anchorage-dependent cell death [84]. In this way, obesity-associated changes in adipokine secretion, by activating corresponding cell-signaling pathways, prepares the cancer cells for metastasis.

Cell adhesion proteins not only physically anchor cells, but also transduce mechanical signals to the actin cytoskeleton in response to the biochemical cues from the microenvironment [84]. Cell adhesion proteins are transmembrane proteins that have three structural segments: the intracellular domain, transmembrane domain, and extra cellular domain [84]. Their cellular arrangement facilitates signal transduction from the extracellular domain to the intracellular cytoplasmic domain [84]. Cadherin is a calcium-dependent transmembrane protein that makes cell–cell contact with adjacent epithelial cells in the extracellular domain [84], while the intracellular domain remains in contact with the actin cytoskeleton. The cytoplasmic domain of cadherin makes contacts with β-catenin, which, in turn, binds to actin binding protein—α catenin [91]. α catenin homodimers compete with Arp 2/3 proteins to bind to the actin filament, and thus suppress actin polymerization [91,92]. Therefore, functionally, cadherin contacts locally regulate actin cytoskeletal homeostasis, which may become impeded during EMT.

The loss of cell polarity is also a hallmark of advanced cancer [93]. EMT plays a role in controlling cell orientation during migration by controlling polarity proteins. Three groups of polarity proteins, namely the Crumbs complex, Par3-Par6-aPKC complex, and Scribble (Scrib)–Disc proteins present in the apical-lateral and basolateral boundaries of epithelial cells, maintaining cell polarity and tissue organization [93]. The abnormal expression of the receptor tyrosine kinase ErbB2 was observed in breast cancer, and its activation causes the dissociation of Par 3 from the Par6–aPKC complex. This ensures the disruption of apical-basal polarity and the induction of EMT [93]. EMT-induced alterations of cell–cell junctions further facilitate the localization of junctional proteins into the leading edge of migrating cells, endowing cells with the migratory capacity [76].

Metastasis requires cancer cells to invade the surrounding tissue to make their way to distant locations. This involves the adherence of cancer cells to the ECM and the proteolytic activity mediated by the matrix metalloproteinase [94]. Matrix metalloproteinases (MMPs) are zinc- and calcium-dependent endopeptidase enzymes responsible for degrading the extracellular matrix and basement membrane proteins [95,96]. Secretory MMPs are released into the extracellular space as inactive zymogen, which becomes activated upon binding to their substrate by disrupting the cysteine–Zn interaction by plasmin, or via oxidative modification by reactive oxygen species (ROS) [95]. Several in vitro studies found that the integrin-mediated activation of MAPK pathways regulate the expression of MMPs through the upregulation of the transcription factor GATA-2 [96,97,98,99]. Clinical studies associated the increased serum concentration of MMPs with breast cancer risk [100,101]. However, their association with obesity or obesity-induced chronic inflammation is still inconclusive. Nevertheless, studies correlated EMT-mediated transcription factor activation with the upregulation of different matrix degradative proteins [81]. For example, Olmeda et al. (2007) observed in an in vitro study that the silencing of Snail1 decreased the expression of the pro-invasive marker MMP9 in breast carcinoma cells [102]. Another possibility involving ECM degradation is that EMT-mediated invadopodium formation can recruit various membrane-tethered proteases (MT-MMP) at the cell matrix contact point to degrade the ECM [81].

Opposing this proteolytic activity are integrin transmembrane proteins, such as cadherin, which function to adhere the cell to the ECM. There are 18 integrin α subunits that heterodimerize with any of the 8 β subunits, making 24 different integrin heterodimers [84]. It has been shown that these different heterodimer combinations trigger distinct responses when bound to the same ligand [94]. Thus, the pattern of integrin expression determines the outcome of the microenvironment influence, and these patterns vary between cancer types [94]. Integrin-mediated cell matrix adhesion serves as a structural anchor point that organizes actin cytoskeletons [103]. Integrin contact with the actin cytoskeleton is mediated by the cytoskeletal linker protein complex talin, vinculin, and α-actinin, via its intracellular domain. The extracellular domain primarily binds with glycoproteins and cell matrix connective tissue components, such as collagen, laminin, and fibronectin [84]. In response to the biochemical cues from the microenvironment, integrin-mediated cell matrix adhesion regulates cytoskeletal arrangements.

### 3.6. Regulatory Proteins in Obesity-Favored Signaling Pathways towards Metastasis

While genetic and epigenetic alterations typically induce cancer initiation, the progression of cancer to the advanced stage is largely influenced by the tumor microenvironment. The hyperactivation of oncogenic proteins and the deletion of tumor suppression proteins, due to genetic mutations, causes an oncogenic transformation in cells [104]. Many of these genetic mutations encode proteins that are components or targets of receptor tyrosine kinase (RTK) proteins controlled by PI3K-Akt and Ras-ERK cell-signaling pathways, such as the epidermal growth factor receptor (EGFR), small GTPase (e.g., Ras), serine/theonine kinases (e.g., Raf and Akt), cytoplasmic tyrosine kinase (e.g., Src), lipid kinase (e.g., phosphoinositide 3-kinase, PI3K), as well as nuclear receptors (e.g., the estrogen receptor, ER) [103]. Under normal physiological conditions, both of the pathways control numerous physiological responses—keeping cell growth and proliferation in check, controlling stress signals, such as cellular apoptosis and DNA damage in response to growth factors and cytokine signaling or ligand binding [104]. However, genetic alterations induce a constant activation of the proteins in these pathways, even in the absence of growth factor signaling. For example, Ras-ERK and PI3K-Akt cell-signaling pathways control estrogen receptor (ER)-dependent tumor cell survival, growth, and proliferation [104]. In the case of obesity, this already challenged cellular ecosystem is further derailed with reciprocal paracrine and juxtaparacrine interactions of non-neoplastic cells inhabiting the microenvironment [104]. Adipocyte-secreted factors may be critical in connecting obesity and the advanced progression of breast cancer via transducing signals from dysfunctional adipocytes to the proximal breast cancer cells [105]. Paracrine interaction of adipokines with the cell-surface receptors on breast cancer cells, followed by the sustained activation of PI3K-Akt and Ras-ERK cell-signaling pathways, influence the metastatic behavior in breast cancer cells. To date, in vitro and in vivo studies have confirmed that obese or dysfunctional white adipocytes stimulate breast cancer progression by increasing the survival, growth, proliferation, migration, and invasion of breast cancer cells [69,83,106,107,108,109]. Other studies reported the direct influence of secreted factors on breast cancer progression. For example, several in vitro studies have shown IL-6 to promote breast cancer cells’ EMT transition, migration, and invasion. Additionally, leptin was reported to enhance cell viability, proliferation, metastatic potential, and stemness [110,111,112]. However, the paracrine effect of those changes on proteins that regulate the biological phenomenon of metastasis in breast cancer cells remains understudied.

To initiate their movement, cells must break off their adhesion contact with other cells and the matrix and form a protrusion in the direction of the movement. At the same time, the forward protrusion contacts the matrix as the rear portion of the cells detaches from the matrix through actomyosin contraction, pushing the cytoplasm forward in the direction of movement. The events that instigate this biological phenomenon are actin polymerization, actomyosin contraction, and cell matrix adhesion [113,114]. The Rho family of GTPase proteins (RhoA, Rac1, and Cdc42) controls these fundamental physiological processes by activating the downstream effector proteins of multiple cells-signaling pathways, including actin regulators, adapter proteins, protein kinases, and phospholipases [115,116]. By controlling those proteins, Rho family GTPase proteins influence normal cellular functions, such as cell adhesion, migration, and invasion [117]. The activation of Rho family GTPase proteins is controlled by the guanine nucleotide exchange factor (GEF) [118]. The molecular switch between the GTP and GDP bound form of Rho GTPase is catalyzed by GEFs, which displaces GDP and allows the binding of GTP on the active site of Rho family GTPase [117]. GTP binding then ensures conformational changes in small GTPases, which enables them to interact with downstream effector proteins [117]. The intracellular signal transducer proteins, Src and PI3K, work upstream of GEF and control its activation in response to the activation of RTK receptors in the presence of growth factors, cytokines and chemokines [119]. Cytokine receptors of the immunoglobulin (Ig) superfamily and GPCR-type receptor can also activate GEFs in response to extracellular stimuli, and can coordinate with the Rho family GTPases’ activation signaling [120].

While GEFs activate the small GTPases, there are two other regulatory proteins that direct their inactivation. GAP (GTPase-activating protein) is one of the regulatory proteins that induces the inactivation of small GTPases by hydrolyzing the bound GTP and switching their confirmation to the previous GDP-bound state [117]. Guanine nucleotide dissociation inhibitors (GDIs) are another set of regulatory proteins that inactivate small GTPase proteins by sequestering them in the cytoplasm and inhibiting their localization in the plasma membrane, mediated by masking their C-terminal lipid moieties [117]. Hence, any disruption to this regulatory activity increases the progression of cancer. Rho family GTPase proteins were observed to be overexpressed in cancer, including breast cancer [117,121]. Their overexpression in cancer implies that there is a constant activation of GEFs in comparison to GAPs and GDIs. This further confirms the importance of the detrimental effect of the obese tumor microenvironment in the progression of cancer.

Another emerging area of research is associating the expression of focal adhesion kinase (FAK) as an upstream regulator of Rho family small GTPase. Integrin-mediated contact of epithelial cells with the ECM can transduce extracellular signals by activating the intracellular tyrosine kinase protein, FAK [122]. FAK is a non-receptor, cytoplasmic tyrosine kinase protein, responsible for transducing signals from integrins in contact with the ECM to intracellular domains [123]. The binding of extracellular ligands (i.e., growth factors, cytokines, and ECM components) with their respective receptor in breast cancer cells, results in the initiation of the intracellular signaling cascade via the activation of FAK (Figure 3). Thus, in addition to integrins, other cell-surface transmembrane receptors, growth factor receptors, cytokine receptors, and G protein-coupled receptors relay extracellular signals to FAKs [124]. Upon activation, FAK undergoes autophosphorylation, creating a binding site for tyrosine kinase proteins, such as Src and PI3K [122]. Src then maximizes the kinase activity of FAK by phosphorylating other tyrosine residues [103]. This dual kinase complex controls the activation of numerous downstream effector proteins, including GEFs and GAPs, which, in turn, regulate the activation of several members of Rho family GTPase proteins [103]. The activation of Rac1 by the Dock180-ELMO1 complex and Cdc42 via Pak-interacting exchange factor-beta (β-Pix) are one of several GEFs that facilitate membrane protrusion formation [103]. In addition, FAK-Srk induces the transient suppression of the RhoA protein through its interaction withp190RhoGAP, to release the cytoskeletal tension during cell spreading [103]. The overexpression of FAK and integrin is the hallmark of some cancers. According to Tai and his colleagues, the growth factor-mediated stimulation of FAK from the tumor microenvironment is critical for cancer progression [125]. The altered expression of growth factors and cytokines associated with the obese tumor microenvironment can facilitates different biological metastatic phenomena, following the activation of Rho family GTPases through FAK-mediated action.

## 4. Conclusions and Future Directions

Breast cancer is a leading cause of cancer-related death in women. While the surgical resection of a localized tumor might be curative, metastatic tumors demonstrate a very poor prognosis for survival. The breast tumor microenvironment has been recognized as a critical contributor to cancer progression and treatment resistance. The situation becomes more devastating in the case of obesity, a metabolic disorder. WAT is the central regulator of whole-body energy homeostasis by acting as an energy reservoir as well as by secreting various bioactive adipokines. With the concurrent rise in glucose, insulin, and lipids after food intake, WAT, actively participates in the process of glucose uptake, triglyceride formation, and insulin sensitization [126]. Conversely, during the fasting stage and exercise, fatty acids are released from the reservoir of WAT and are oxidized by the muscles and liver to generate fuel for the body [126]. This increase in the oxidation of fat is mediated by the activation of the AMP-activated protein kinase (AMPK) signaling pathway. This pathway also increases the uptake of circulating glucose in the muscles and liver, thereby reducing insulin resistance [127]. Hence, an absence of fat mass in the body causes hyperglycemia, hyperlipidemia, insulin resistance, and fatty liver [128,129]. Additionally, the presence of immune cells in healthy breast tissue maintains tissue homeostasis and an immune suppressive microenvironment by keeping track of apoptotic cells, angiogenesis, and ECM remodeling [31,127]. However, chronic overnutrition, followed by the overaccumulation of fat in WAT, hampers the energy homeostasis and triggers a local and systemic inflammatory response (as described above). In lean mass breast cancer patients, the normal function of WAT is retained, maintaining the body’s energy homeostasis. However, in obese patients, during fasting or due to the activation of the lipolytic pathway, the release of FFA is increased, fueling the adjacent breast cancer cell’s metabolic need. In vitro studies addressed how cancer cells reprogram the adipocytes to cancer-associated adipocytes and use adipocyte-derived free fatty acids for metabolic purposes through beta oxidation [25,69,70,130]. Moreover, in situ studies have shown how chronic disease conditions, such as obesity and diabetes, impact breast adipocytes to facilitate the growth and survival rate of breast cancer cells [69,131]. In lean mass individuals, immune cells residing in the adipose tissue are predominantly immunosuppressive and secrete anti-inflammatory cytokines, such as IL-10 and IL-4 [132]. However, in obese individuals, due to adipocyte dysfunction and low-grade inflammation, the ratio of pro-inflammatory cytokines, such as TNF-α, IL-6, and IL-8, exceeds the levels of anti-inflammatory cytokines to mitigate, thereby inducing a pro-inflammatory microenvironment [129]. This, together with the altered adipokine secretions from WAT in obesity, (i.e., the decreased secretion of adiponectin, increased secretion of leptin, IL-6, resistin, plasminogen activator inhibitor type 1, and monocyte chemotactic protein-1 (MCP-1) [126,127]) activates cell-signaling pathways in breast cancer cells that are correlated with growth, proliferation, EMT, migration, and invasion [69,83,106,107,108]. As discussed above, adipokines, such as leptin, interacts with the leptin receptor (ObR) to drive EMT [133], facilitating migration by interacting with Rho family GTPases, and invasion by the activation of MMPs. Furthermore, the knockdown of leptin/ObR signaling facilitates an immunosuppressive microenvironment and redirects tumor growth to distant sites [134].

Undoubtedly, adiposity in obesity is the key influencer associated with the adverse progression of breast cancer. However, one clinical report showed increased numbers of cases of breast cancer recurrence concurrent with decreased survival in normal BMI breast cancer patients (BMI < 25 kg/m^2^) with stage I and II disease, in comparison to obese or overweight BMI (BMI > 25 kg/m^2^) patients [135]. Furthermore, another clinical report highlighted all the research studies demonstrating high BMI subjects with markedly improved immune-efficacy responses, following immune checkpoint blockade treatment in comparison to normal BMI subjects, despite the contradictory role of obesity in immunosuppression [136]. With such obesity paradox in cancer, it is difficult to decide the exact role of obesity-associated abnormalities in cancer progression. However, BMI is not an accurate index to compare lean mass and obese mass individuals [137]. It fails to account for the variability in body composition with respect to age, sex, and ethnicity. Additionally, the presence of confounding variables, such as access to food, socioeconomic status, physical activity, diet, alcohol consumption, smoking status, the usage of hormonal drugs and others, or reverse casualties (e.g., loss of weight in obese patients due to illness), further leads us towards a misinterpretation of data associated with BMI [137]. Moreover, some clinical variables, such as comorbidity and the effect of the microbiomes, also need to be considered to fully exploit and optimize the effects of obesity, represented with a BMI index [136]. There is still no standardized alternative measure of body composition that considers age, sex, and the existence of varying ethnicities [137]. Hence, a well-planned clinical research study design with the proper timing of BMI determination (pre, peri, and post diagnosis as well as treatment), accounting for all the relevant covariables mentioned above, needs to be considered to find the answers for the relationship of obesity in carcinogenesis and cancer progression [135,137].

Additionally, despite the significant advancements in basic biology research linking adipose tissue biology and breast cancer, questions remain unanswered concerning the coordinated effect of adipocyte-secreted proteins with the presence or absence of tumor-associated macrophages in activating cell-signaling pathways related to metastasis and immune evasion. Unfortunately, there are even less studies dissecting how relevant proteins or cell-signaling pathways contribute to those biological phenomena. A precise coordination and integration of signals from adipocytes to key proteins in breast cancer cells might be one of the underlying factors connecting obesity and the advanced progression of breast cancer. Understanding this interactive biology between breast cancer cells and adipocytes can introduce us to new metabolic and cell-signaling target molecules that reveal the obesity-associated breast cancer progression relationship. This can open new treatment avenues for more effective approaches to breast cancer in obese, postmenopausal women. Until then, adopting lifestyle interventions (i.e., regular exercise; avoiding high-carbohydrate and high-calorie diets; increased consumption of fruits, vegetables, and other fibrous diets; and, most importantly, reducing stress) continues to be the most effective strategy to reduce the risk of obesity-induced breast cancer.

## Figures and Tables

**Figure 1 ijms-23-02480-f001:**
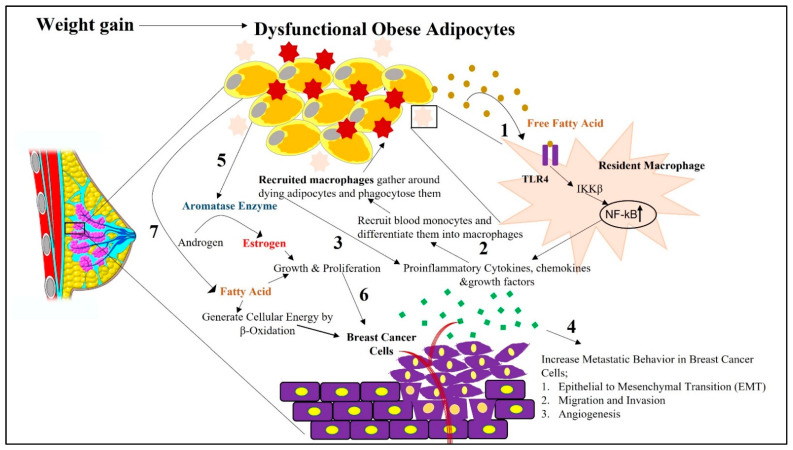
Schematic representation depicting how obesity-associated changes in white adipocytes adjacent to breast cells in the tumor microenvironment can impact breast cancer initiation and its progression. Impaired physiological function of white adipocytes due to increased fat accumulation, followed by hypoxia and ECM stiffness, causes them to undergo apoptosis-induced cell death. **1.** During this cell death phase, cellular contents are released from dying adipocytes. These released contents, such as free fatty acids, cause receptor-mediated activation of TLR4 in resident macrophages. This activation of TLR4 stimulates the secretion of proinflammatory cytokines, chemokines, and growth factors from the resident macrophages via NF-kB activation. **2.** From the released chemokines, CCL2 facilitates the recruitment of blood monocytes around the dying adipocytes from the surrounding intravascular spaces. In the presence of environmental cues in the breast tumor microenvironment, these monocytes differentiate into macrophages, which are then considered recruited macrophages. **3.** There is further increased intracellular signaling in macrophages via NF-kb-, STAT3-, and JNK-related pathways, followed by the release of proinflammatory cytokines, creating a state of chronic inflammation. **4.** Increased secretion of proinflammatory cytokines and hormones from white adipocytes, further facilitates the metastatic progression of breast cancer via their paracrine influence on proximal breast cells. **5.** Obesity-associated increased release of proinflammatory cytokines further increases the expression of aromatase in white adipocytes, which then converts androgens to estrogens in adipose tissues. **6.** This promotes mammary tumorigenesis by increasing the growth and proliferation of breast cancer cells. **7.** Obese adipocyte-released free fatty acids shunt breast cancer cells towards β-oxidation as a source of energy to sustain breast cancer progression.

**Figure 2 ijms-23-02480-f002:**
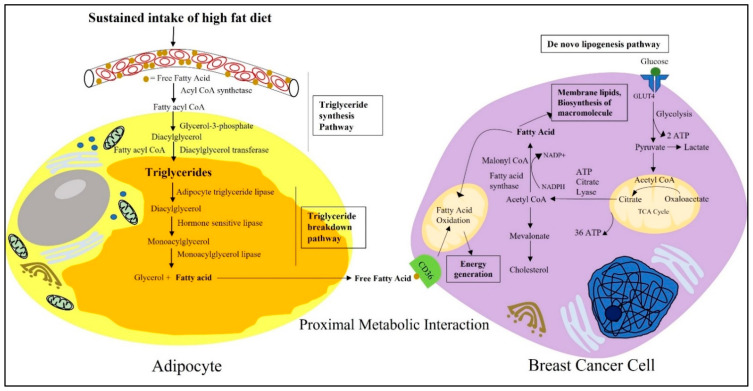
Schematic representation depicting the proximal metabolic interactions between adipocytes and breast cancer cells. Fatty acids are the unit molecules of fat, present in excess in the blood plasma of obese individuals. As free FAs can be toxic at higher concentrations, they are stored in the cytosolic lipid droplets of white adipocytes in the form of triglycerides following the triglyceride synthesis pathway. Due to excess energy demands and metabolic requirements, breast cancer cells can mobilize these stored fats from surrounding adipocytes and use them to compensate for their energy needs. Breast cancer cells use the fat in the form of fatty acids that are derived from the triglycerides stored in white adipocytes. Free fatty acids released from white adipocytes are then taken up by breast cancer cells through the transmembrane channel protein CD36A present on its cell surface. Apart from that, breast cancer cells are also able to generate fatty acids from blood-derived glucose following de novo lipogenesis pathways. Breast cancer cells then use the unit molecule of fatty acids to fulfill their ever-expanding metabolic needs, to synthesize new macromolecules and membrane lipids.

**Figure 3 ijms-23-02480-f003:**
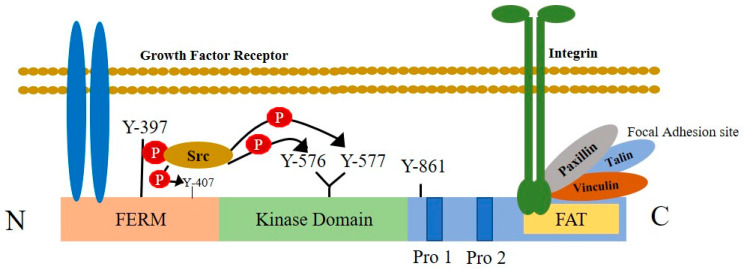
Structural features of focal adhesion kinase and its activation by cell-surface receptors. FAK is composed of a central kinase domain bordered by the N-terminal FERM homology domain and C-terminal region containing two proline-rich motifs and a FAT domain. The interaction of integrins with extracellular ligands, increases large macromolecular clusters on the cell cytoplasmic side that anchors the actin cytoskeleton to the plasma membrane in connection with the integrin-associated proteins talin, paxillin, and vinculin. This is known as the focal adhesion site. FAK connects with the focal adhesion sites of integrin through its C-terminal domain containing a focal adhesion targeting (FAT) sequence. The N-terminal FERM domain integrates signals from growth factor receptors. In response to extracellular stimuli, FAK activation causes the autophosphorylation of FAK at the Tyr397 residue and creates an SH2 domain docking site that interacts with proteins, such as Src. This interaction activates the Src tyrosine kinase, which further trans-phosphorylates other tyrosine residues placed on FAK and maximizes kinase activity.

**Table 1 ijms-23-02480-t001:** Distinct features of breast cancer based on molecular histotypes for U.S. women.

Molecular Subtypes	Luminal A	Luminal B	HER2 Positive	Triple Negative	Inflammatory Breast Cancer
% of Breast Cancers (U.S. Women) [12]	11%	73%	4%	12%	Unknown
Receptor Expression	ER+ PR+ HER2+	ER+ PR± HER2-	ER- PR- HER2+	ER- PR- HER2-	ER-, PR-, HER2+ TNBC
Histological Grade	Low	Intermediate	High	High	High
Prognosis	Good	Intermediate	Poor	Poor	Poor
Ki67 by IHC	Low	High	High	High	High
Response to Treatment	Endocrine therapy: anti-estrogen aromatase inhibitors—Anastrozole Exemestrozole Letrozole	Endocrine therapy: anti-estrogen aromatase inhibitors—Anastrozole Exemestrozole Letrozole	HER2-targeted drugs: Trastuzumab Pertuzumab Neratinib	Treatment: Taxanes— Paclitaxel Docetaxel Anthracyclines Doxorubicon 5-flurouracil	Treatment: HER2+ targeted therapy

## Data Availability

Not applicable.

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
