# Peer review of "The Complex Biology of the Obesity-Induced, Metastasis-Promoting Tumor Microenvironment in Breast Cancer"

_ijms, 2022, doi:10.3390/ijms23052480_

Round 1

Reviewer 1 Report

over all a very good review 

needs minor revision for example in the table you mentioned inflammatory breast cancer as a HER-2 , it does not need to be HER-2 to have inflammatory breast cancer , it can happens with other types particularly triple negative

Author Response

Thank you for the review and and pointing out our omission that TNBC are strongly associated with inflammatory breast cancers. We have corrected that oversight in Table 1.

Reviewer 2 Report

Authors present a work addressing: ‘The complex biology of the metastasis-permissive obese tumor microenvironment in breast cancer’. Authors discussed the complex biology of obesity-induced changes in white adipocytes to better understand the relationship between obesity and breast cancer progression to metastasis.

The topic of the article is relevant for clinical practice. However, the paper presents a few minor and major issues including:

  1. If this article has been sent to worldwide publisher the epidemiological data should be extended on global data.
  2. Authors should provide clear aim of the review.
  3. Authors should provide methodology i.e. the number of the analysed articles?, which databases authors used?, what did algorithm the authors use.
  4. Figures no 1 and 2 should be improved by increasing the size of description.
  5. In my opinion, Authors should add limitation of their literature analysis.
  6. Authors should discuss and add following articles:

a) Bielawski K, Rhone P, Bulsa M, Ruszkowska-Ciastek B. Pre-Operative Combination of Normal BMI with Elevated YKL-40 and Leptin but Lower Adiponectin Level Is Linked to a Higher Risk of Breast Cancer Relapse: A Report of Four-Year Follow-Up Study. J Clin Med. 2020 Jun 4;9(6):1742.

b) Lennon, H.; Sperrin, M.; Badrick, E.; Renehan, A.G. The obesity paradox in cancer: a review. Curr. Oncol. Rep. 2016, 18, 56.

c) Murphy, W.J.; Longo, D.L. The surprisingly positive association between obesity and cancer immunotherapy efficacy. JAMA 2019, 321, 1247– 1248.

d) Sánchez-Jiménez, F.; Pérez-Pérez, A.; de la Cruz-Merino, L.; Sánchez-Margalet, V. Obesity and breast cancer: Role of leptin. Front Oncol. 2019, 9, 596.

Author Response

Dear Reviewer 2.

Thank you for your careful and constructive review of our manuscript. Below we address the changes requested.

  1. If this article has been sent to worldwide publisher the epidemiological data should be extended on global data.

We have included global statistics as well as extended epidemiological data in a revised chapter 2.1 highlighted in green and in section 2.2

  1. Authors should provide clear aim of the review.

In the abstract and in the introduction - highlighted in green

  1. Authors should provide methodology i.e. the number of the analysed articles?, which databases authors used?, what did algorithm the authors use.

We have addressed these critiques in a new Methodology section at the end of the review. However, we point out that no metadata analysis was used, rather a comprehensive review of research articles from  PubMed, Google Scholar, ScienceDirect, Web of Science etc. Another arguable limitation is that key words or dates for the inclusion and exclusion of articles, while searching the databases for relevant information.

  1. Figures no 1 and 2 should be improved by increasing the size of description.

Thank you for your comments. We have increased size of figure and fonts therein.

  1. In my opinion, Authors should add limitation of their literature analysis.

Added to the methodology section. Please see response above.

  1. Authors should discuss and add following articles:

a) Bielawski K, Rhone P, Bulsa M, Ruszkowska-Ciastek B. Pre-Operative Combination of Normal BMI with Elevated YKL-40 and Leptin but Lower Adiponectin Level Is Linked to a Higher Risk of Breast Cancer Relapse: A Report of Four-Year Follow-Up Study. J Clin Med. 2020 Jun 4;9(6):1742.

b) Lennon, H.; Sperrin, M.; Badrick, E.; Renehan, A.G. The obesity paradox in cancer: a review. Curr. Oncol. Rep. 2016, 18, 56.

c) Murphy, W.J.; Longo, D.L. The surprisingly positive association between obesity and cancer immunotherapy efficacy. JAMA 2019, 321, 1247– 1248.

d) Sánchez-Jiménez, F.; Pérez-Pérez, A.; de la Cruz-Merino, L.; Sánchez-Margalet, V. Obesity and breast cancer: Role of leptin. Front Oncol. 2019, 9, 596.

Thank you for referring us to these articles. We have read and incorporated them into our review and it has strengthened the review.

Reviewer 3 Report

Title: "The Complex Biology of the Metastasis-Permissive Obese Tumor Microenvironment in Breast Cancer” 

Authors: Noshin Mubtasim, Naima Moustaid-Moussa3 and Lauren Gollahon

Comments:

In this work, Mubtasim et al. discuss the complex biology of obesity-related changes in white adipose cells to better understand the relationship between obesity and breast cancer progression to metastasis. interesting topic structurally and linguistically well summarized; however, there are some mistakes of carelessness:

Major points:

1: The title is worded unfortunate and could be worded more understandable. I think it should rather be (as also described in the abstract) "obesity-induced TME", because TME itself is not "obese"

Suggestion: The complex biology of the metastasis-promoting obesity-induced tumor microenvironment in breast cancer.

2: I don't like chapter 1 so much in terms of structure. I would rather do a general introduction as "1." then a second chapter with the more detailed subchapters of 1.

3: Page 3, line 88, reference correction: Bobby et al. should read Daly et al. (Ref. 13).4: Page 3-4, Chapter 1.

4: In this chapter, a distinction should be made between an in vitro TME of basic research and the in vivo TME found in the diseased patient. Both should be mentioned and the similarities highlighted.

5: Page 5-16, Chapter 2: Throughout Chapter 2, it should be clarified which statements refer to in vitro tests and which results are from patient studies.

6: Page 16, lines 696-700: What possible new treatment strategies do the researchers foresee? The most important conclusion is that a healthy lifestyle is important, which is already known.

7: List of abbreviations would be desirable (a lot of abbreviations are used).

8: Some abbreviations are introduced several times, e.g:-page 7 line 262: CSF1-page 9 line 357: TCA-page 10 line 385: TAG-page 14 line 606: FAK-page 15 line 661: ECM

9: Page 10 line 411: PGE2 is written out here; the abbreviation PGE2 has already been introduced previously

10: Page 3 lines 87-89: In this sentence I would briefly mention why this is so.

11: Page 3 line 90: It says that African Americans are diagnosed with breast cancer more often than white women; but line 88 still says that white women have the highest incidence - that sounds contradictory?

12: Page 6 Figure 1: it would be helpful if the numbers in the picture were a bit bigger/clearer to find them faster

13: Partial spelling errors, e.g.:-Page 4 lines 142-144 + page 15 line 644: the plural of man and woman should be "men" and "women" respectively.-Page 6, line 240: it should be "act" (because plural).-Page 8, line 317: patients (plural).

14: Page 2 Table 1: To which countries do the data shown in the table refer? (e.g., % of breast cancer cases) also to the U.S.? Or worldwide? please specify

15: Page 11 line 471: CRC is mentioned here -> are there similar results from breast cancer cell studies?

16: Page 5 chapter 2.1.: partly very similar to 1.5.

17: General review of the form: There are some character/letter errors in the text that I will not itemize here.

Author Response

Dear Reviewer 3.

Thank you for your careful reading and critical review of our manuscript. Below, we respond to those comments.

1: The title is worded unfortunate and could be worded more understandable. I think it should rather be (as also described in the abstract) "obesity-induced TME", because TME itself is not "obese"

Suggestion: The complex biology of the metastasis-promoting obesity-induced tumor microenvironment in breast cancer.

We have changed the title to the suggested example.

2: I don't like chapter 1 so much in terms of structure. I would rather do a general introduction as "1." then a second chapter with the more detailed subchapters of 1.

We have restructured as suggested, including a brief introductory chapter highlighting the problem and importance of breast cancer including statistics and the overall aim of the review. Following this, we begin chapter 2 with Breast cancer keeping the detailed,  itemized subheadings. Each subsequent chapter and subheading has been appropriately renumbered.

3: Page 3, line 88, reference correction: Bobby et al. should read Daly et al. (Ref. 13).4: Page 3-4, Chapter 1.

Corrected - Thank you

4: In this chapter, a distinction should be made between an in vitro TME of basic research and the in vivo TME found in the diseased patient. Both should be mentioned and the similarities highlighted.

This has been addressed in sections 2.3 and 2.4 and are highlighted in yellow. We specifically draw attention to the in vivo TME in these instances as it is relevant to the obesity-induced changes for the review. 

5: Page 5-16, Chapter 2: Throughout Chapter 2, it should be clarified which statements refer to in vitro tests and which results are from patient studies.

Based on the reviewer's suggestion, we have clarified the referred literature as either in vitro (lab conditions), in vivo (rodent models) or in situ - referring to the patient findings. However, the majority of patient studies is included in chapter 4 - Conclusion and Future directions

6: Page 16, lines 696-700: What possible new treatment strategies do the researchers foresee? The most important conclusion is that a healthy lifestyle is important, which is already known.

A new paragraph has been added to address this response in Chapter 4 - Discussion and future directions

7: List of abbreviations would be desirable (a lot of abbreviations are used).

We appreciate the reviewer's observation. A list of abbreviations has been included at the end of the manuscript preceding the references. We have also gone through the review carefully and tried our best to spell out the term first when appropriate and follow this with the abbreviation in parentheses.

8: Some abbreviations are introduced several times, e.g:-page 7 line 262: CSF1-page 9 line 357: TCA-page 10 line 385: TAG-page 14 line 606: FAK-page 15 line 661: ECM

These have been corrected.

9: Page 10 line 411: PGE2 is written out here; the abbreviation PGE2 has already been introduced previously

This has been addressed in the revised manuscript.

10: Page 3 lines 87-89: In this sentence I would briefly mention why this is so.

We have added an explanation for this highlighted in yellow and beginning ~ line 90.

11: Page 3 line 90: It says that African Americans are diagnosed with breast cancer more often than white women; but line 88 still says that white women have the highest incidence - that sounds contradictory?

Thank you for the observation -- it certainly was contradictory. THis error has been corrected.

12: Page 6 Figure 1: it would be helpful if the numbers in the picture were a bit bigger/clearer to find them faster

We have addressed this enlarging the figure and the fonts.

13: Partial spelling errors, e.g.:-Page 4 lines 142-144 + page 15 line 644: the plural of man and woman should be "men" and "women" respectively.-Page 6, line 240: it should be "act" (because plural).-Page 8, line 317: patients (plural).

Corrected, thank you.

14: Page 2 Table 1: To which countries do the data shown in the table refer? (e.g., % of breast cancer cases) also to the U.S.? Or worldwide? please specify

This has been clarified.

15: Page 11 line 471: CRC is mentioned here -> are there similar results from breast cancer cell studies?

Thank you for this question. Yes and the example is included following the colorectal reference.

16: Page 5 chapter 2.1.: partly very similar to 1.5.

Thank you for the comment. While many of the molecules may overlap, the gist of section 1.5 is on the TME whereas 2.1 is more focused on the ECM directly associated with the adipocytes and how these changes may contribute to the TME via adipocyte membrane rigidity and ultimate destruction, spewing the adipocyte contents into the microenvironment.

17: General review of the form: There are some character/letter errors in the text that I will not itemize here.

We apologize for the missed opportunities to fix these errors. We have gone over the manuscript carefully and it is out hope we found these errors.

Round 2

Reviewer 3 Report

The authors have satisfactorily addressed the concerns raised in the original version. The revised version is significantly improved. No further concerns.